# Saltwater Intrusion in the Changjiang River Estuary in Response to the East Route of the South-to-North Water Transfer Project in the New Period after 2003

**Huiming Huang [1,2], Yan Wang [1,*], Sheng Wang [1], Yinyu Lan [3] and Xiantao Huang [2]**

[1] College of Harbour, Coastal and Offshore Engineering, Hohai University, Nanjing 210098, China; sheng.wang@hhu.edu.cn (S.W.)

[2] Key Laboratory of Coastal Disaster and Protection of Ministry of Education, Hohai University, Nanjing 210098, China; hxthhu@gmail.com

[3] Fujian Provincial Key Laboratory of Coast and Island Management Technology Study, Xiamen 350003, China; yinyulanhhu@163.com

\* Correspondence: yanwang@hhu.edu.cn

**Abstract:** The continuous operation of the Three Gorges Reservoir since 2003 has altered the annual runoff into the Changjiang River Estuary, significantly affecting patterns of saltwater intrusion. This has become more pronounced with the development of the East Route of the South-to-North Water Transfer Project, which has changed the runoff distribution and saltwater dynamics once again. Recognizing the critical need to understand these changes, this study employs numerical simulations to investigate the impact of the East Route of the South-to-North Water Transfer Project's water abstraction on saltwater intrusion in the Changjiang River Estuary post-2003. It assesses intrusion distances, freshwater availability, and periods when water intake might be compromised due to high salinity. Our findings indicate that the East Route of the South-to-North Water Transfer Project markedly influences intrusion patterns. By modeling various runoff scenarios, the study delineates the correlation between average monthly runoff at the Datong Hydrological Survey Station and estuary salinity. It then suggests optimal ecological discharge levels to manage saltwater intrusion effectively. This research provides insights which are necessary for informed water management and ecological protection in the region.

**Keywords:** the Changjiang River Estuary; the east route of the south-to-north water transfer project; saltwater intrusion distance; freshwater resource; unsuitable water intake time; ecological discharge





## 1. Introduction

Estuarine saltwater intrusion is one of the most concerning marine disasters. Therefore, estuarine cities must aim to forecast and understand saltwater intrusion to avoid disasters and formulate saltwater intrusion prevention measures.

As China's largest estuary, the Changjiang River Estuary is located beside Shanghai City, which is also the largest city in China. Saltwater intrusion in the Changjiang River Estuary is a long-standing natural hydrological phenomenon caused by tidal activity. It usually occurs during dry periods, when salt tides form as a result of seawater backing up into the estuary. Therefore, the saltwater intrusion in the Changjiang River Estuary has attracted much attention for decades. Over the years, many studies have been carried out on the characteristics and laws of saltwater intrusion in the Changjiang River Estuary. In usual, it can be concluded that the saltwater intrusion is mainly restricted by the following factors:

(1) Estuarine topography. The pattern of the three levels of bifurcation and the four sea outlets in the Yangtze River estuary (Figure 1) determine the complex and changing characteristics of runoff tropism and tidal dynamics, which strengthens the influence of estuarine topography on saltwater intrusion. Then, natural evolution or human activities

erode and silt the topography or shoreline, resulting in changes in the width of the river channel and the resistance of the riverbed, causing changes in the runoff top flush and tidal dynamics, and further causing the evolution of saltwater intrusion in the estuary [1–5].

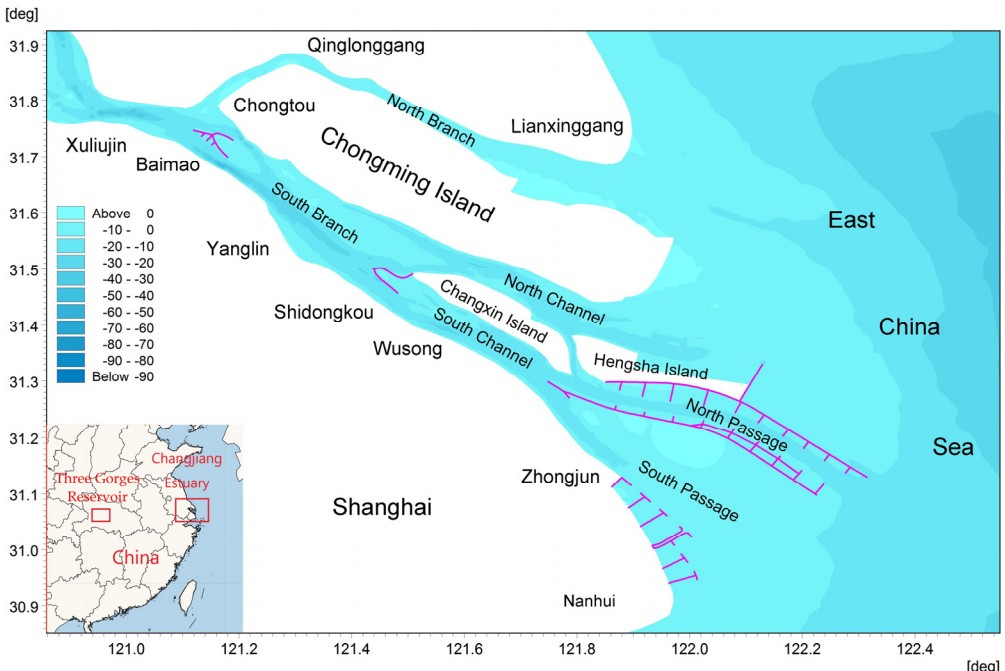

**Figure 1.** River regime of the Changjiang River Estuary.

(2) Estuarine hydrodynamics. The hydrodynamic environment of the Changjiang River Estuary is generally complex. Common estuarine hydrodynamic factors such as tidal power, wind, waves and sea level rise may all have impacts on saltwater intrusion. It is generally believed that tidal power dominates saltwater intrusion in the estuary, and the specific wind and wave environments also cause changes in saltwater intrusion, while sea level rise determines the long-term trends in the evolution of saltwater intrusion [6–11]. Rising sea levels have intensified seawater intrusion, resulting in increased salinity and even excessive chloride levels, making freshwater aquifers more vulnerable [12].

(3) Runoff conditions from river basin. With the changes in the climate environment and the impact of human activities, the runoff in the Yangtze River Basin is also changing, thus further causing changes in saltwater intrusion in the estuary [10,13–17].

Indeed, in recent years, human activities have played an increasing role in saltwater intrusion in the estuary. After the opening of the Three Gorges Reservoir in 2003, the runoff and sediment conditions in the Yangtze River estuary have changed considerably. As shown in the daily discharge process at the Datong hydrographic station from 1950 to 2016 (Figure 2), a discharge flow of less than 10,000 m$^3$/s existed at the Datong hydrographic station in almost every dry-water period before 2003, but such a situation is almost impossible to find in any dry-water period after 2003. The intrusion of salty water into the estuary of the Yangtze River during the dry-water periods after 2003 has also been significantly changed. Therefore, the period after 2003 is regarded as a new period in the study of salt tides in the Yangtze River Estuary.

Nevertheless, with the planning and implementation of the East Route of South-to-North Water Transfer Project with the water intake located in the Yangzhou City (Figure 3), which will pump runoff from the Changjiang River at about 500 m$^3$/s in Phase I and 1000 m$^3$/s in Phase II, the runoff into the Changjiang River Estuary will inevitably decrease in the dry season. Consequently, the beneficial effect of the increased runoff into the estuary from the Three Gorges Reservoir in the dry season on the prevention of saltwater intrusion in the Changjiang River Estuary will be weakened or even completely eliminated. For

example, salinity remained practically constant at around 0.1‰ during the July 2010 flood. During the dry period in January 2011, saline intrusion occurred with a maximum salinity 150 times that of the flood period [18]. Therefore, the mechanisms of saltwater intrusion and engineering measures for the prevention saltwater intrusion in the estuary during the dry seasons since the opening of the Three Gorges Reservoir may also change [19–21].

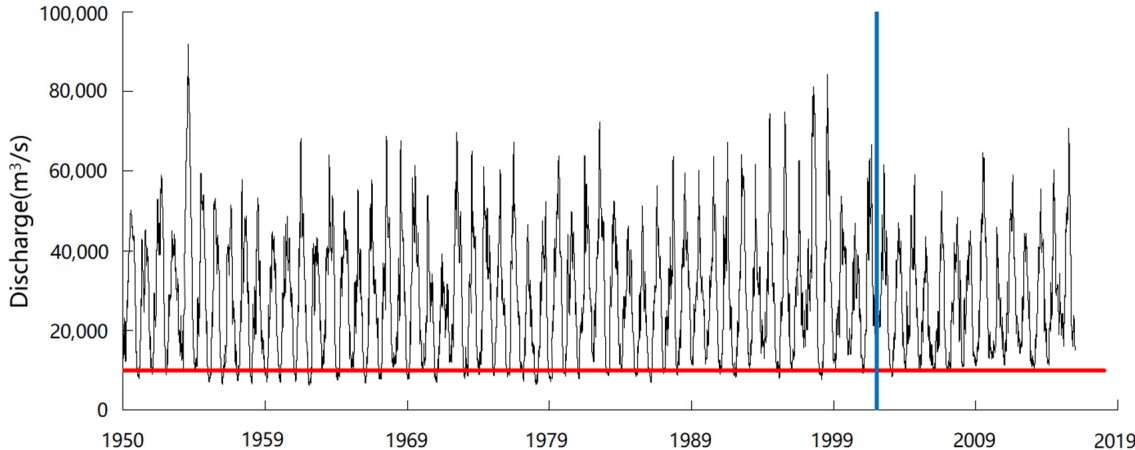

**Figure 2.** Datong Hydrological Survey Station daily discharge during 1950–2016 (Black line is Datong Hydrological Survey Station daily discharge, red line is Datong Hydrological Survey Station discharge as 10,000 m$^3$/s, blue line divides before and after 2003).

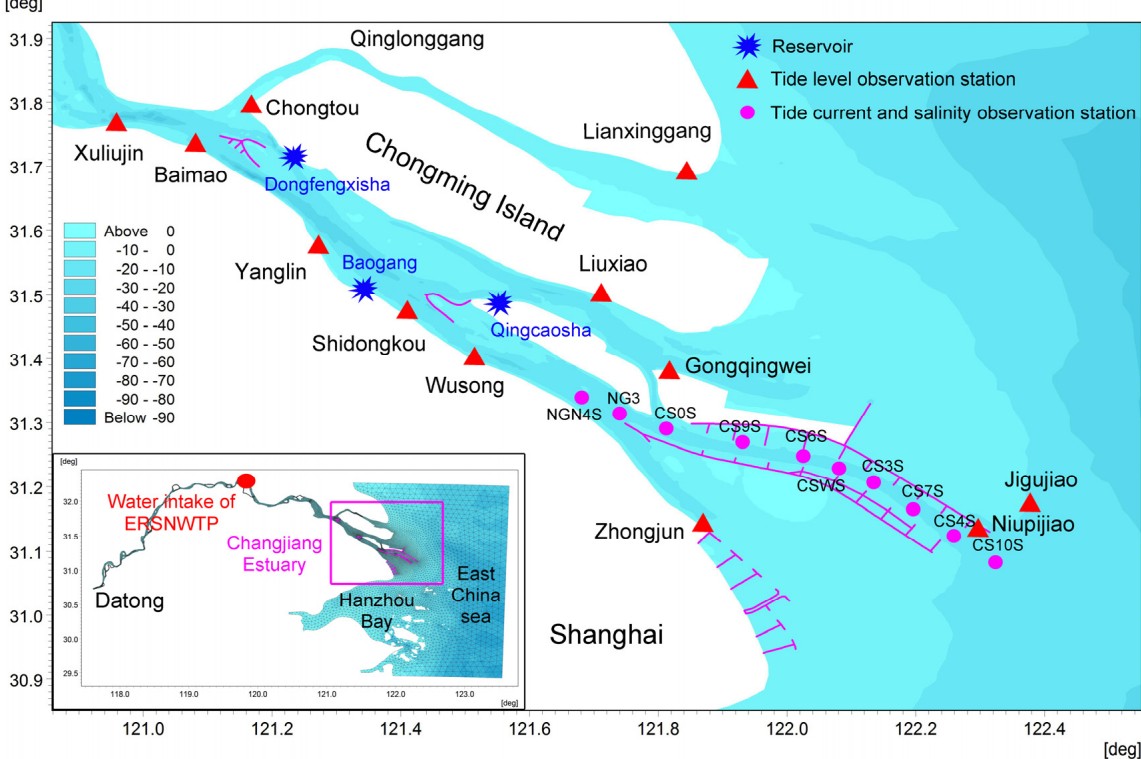

**Figure 3.** Schematic of model range, grid subdivision and locations of hydrological stations.

For this reason, based on the background of the new period after 2003, this paper studies the characteristics and mechanisms of saltwater intrusion in the Changjiang River Estuary under the action of Phase I and Phase II of the East Route of the South-to-North Water Transfer Project.

## 2. Materials and Methods

### 2.1. Methodology and Data

In general, saltwater intrusion has three dimensional characteristics; thus, a three-dimensional numerical model created with MIKE21 (version 2014) software (https://www.dhigroup.com/ (accessed on 12 August 2023)) is used here. Numerical modeling of salinity in three-dimensional flows was produced using the MIKE21 Flow Model FM unstructured grid hydrodynamic (HD) module and its attached thermo-saline transport (TS) module. On the basis of good validation, the spatial and temporal variations in the flow field and salinity in the Changjiang Estuary during the dry season are analyzed in detail, and preliminary calculations are made for the process of saline backwater in the North Branch and the backwater salt flux. The spatial discretization in the MIKE21 model is solved with the finite volume method, and the temporal discretization of the model is solved with the first-order explicit Euler method in order to reduce the computation time due to the large model range. To simulate hydrodynamic and saltwater intrusion, this paper uses a simulation approach. The simulation process takes into account changes in the density of the water body under the conditions of saltwater intrusion, thus the model results have been enhanced.

The data are measured data of synchronous full tidal flow salinity in the Changjiang River Estuary waters, and model calculations are based on China's 1985 National Elevation Datum. The tidal and runoff conditions were chosen based on an analysis of past hydrological data. The hydrographic data used for model validation were synoptic tide levels at tidal stations and stratified current speed, direction and salinity data from fixed vertical stations during a complete spring, middle and neap tide period in March 2016. The specific station locations are shown in Figure 3. The land boundary used in the model adopts the actual landline of the coast and river. And the topography of the model is spliced with underwater topographic data collected from different regions between 2002 and 2017.

The Changjiang Estuary is a large estuary, and the influence of wind cannot be disregarded [22,23]. Therefore, the model takes into account the driving effect of the wind in the hydrodynamic simulation process. The model uses hindcast data for the wind field, and the multi-year monthly average wind field data for January and February, during the dry season, were used as the input wind field data to drive the model (https://www.ecmwf.int/ (accessed on 26 August 2023)), time step is 6 h.

### 2.2. Numerical Model

#### 2.2.1. Model Setting

In order to describe the combined effects of runoff and tide power in the estuary, the east boundary of the model is set near 123.5° E, the north boundary is near 32.3° N, and the south boundary is near 29.5° N (Figure 3). The west riverine boundary of the model is located at Datong Hydrological Survey Station, which is the tidal zone of, or the boundary of the tidal reach of, the Changjiang River, wherein the three open-sea boundaries are driven by the global tidal wave model [24]. The riverine boundary is derived from measured discharge data from the Datong Hydrological Survey Station. The open boundary conditions for salinity in the outer sea are given by the full tidal synoptic measurements, the boundary salinity is linearly interpolated at each computational time step in the model and the boundary salinity is finally determined from the radial boundary conditions based on the inflow and outflow of the water body at the boundary.

The simulation domain uses triangular grids for plane subdivision. The grid outside the Changjiang River Estuary is bigger, and the grid inside the Changjiang River Estuary and tidal reach is smaller. The grid resolution is between 20–3000 m and locally encrypted in the North Branch and South Branch, North Channel and South Channel and the North Passage and South Passage (Figure 3). The number of grids is 75,119 and the number of grid nodes is 40,091.

For a sensitivity analysis of the modeled Koch forces, vertical layering was performed, and the results showed that the best results were obtained in the presence of Koch forces and were in 11 layers, which are, respectively, 0.05, 0.1, 0.1, 0.1, 0.1, 0.1, 0.1, 0.1, 0.1, 0.1,

0.05 times the water depth. The time step was set as 3 s. The bottom roughness was taken to be between 0.01 and 0.02 after repeated adjustments according to the water depth and bedform in the model area. The k-ε model was used to calculate the coefficient of eddy viscosity for the water flow, and the salinity diffusion followed the salinity transport equation [25–27]. k-ε is the most widely used turbulence model in engineering calculations. The horizontal diffusion coefficient of salinity is mainly caused by turbulence in the water flow and is considered to be proportional to the eddy viscosity coefficient of the water flow with a ratio of 1. The model is simulated using cold start. The initial water level field and flow velocity field are taken to be zero, and any deviation in the initial conditions of the water flow will disappear quickly under the control of the boundary conditions. The intra-estuary is obtained by interpolating the data from several real measurements in the dry season.

The article simulates the salinity for two months, with the first month being stabilization time for the numerical simulation and the second month allowing for the stabilization of the salinity results for research and analysis.

The model range is shown in Figure 3 and grids are shown in a sub-figure.

2.2.2. Model Verification

In order to quantitatively analyze the accuracy of each simulated parameter at all stations, the correlation coefficient (CC), the commonly used test index skill score (SS) and root mean square error (RMSE) are adopted for comprehensive evaluation; the larger values indicate that the model's simulated results are approximately accurate. SS is evaluated according to Allen's classification method [28]. Their specific expressions are as follows:

$$CC = \frac{\sum (X_{mod} - \overline{X}_{mod})(X_{obs} - \overline{X}_{obs})}{\sqrt{\sum (X_{mod} - \overline{X}_{mod})^2 \sum (X_{obs} - \overline{X}_{obs})^2}} \tag{1}$$

$$SS = 1 - \frac{\sum (X_{mod} - X_{obs})^2}{\sum (X_{obs} - \overline{X}_{obs})^2} \tag{2}$$

$$RMSE = \sqrt{\frac{\sum (X_{obs} - X_{mod})^2}{n}} \tag{3}$$

where X is, respectively, tidal level, current speed and direction, salinity; superscript represents average value; subscript mod and obs are, respectively, model result and observation result.

When SS < 0.2, the simulated result is poor, and while 0.2 < SS < 0.5, the simulated result is good; when 0.5 < SS < 0.65, the simulated result is very good, and when SS > 0.65, the simulated result is excellent.

The average values of the CC and SS for all observation stations during the simulation duration in dry season are shown in Table 1.

As Table 1 shows, all RMSEs met the error requirements, and the CC and SS values for the tidal level all exceed 0.90, indicating that the tidal level simulation results are excellent. Meanwhile, the CC and SS values for current speed and direction are between 0.76 and 0.87, indicating that the simulation results are also excellent. The CC and SS values for salinity decrease a little compared with those of other elements, but still remain above 0.65, indicating that the simulation results of salinity are still excellent.

In summary, the simulated results for tidal level, layered current speed, direction and salinity are all in good agreement with the observed values. The verification accuracy of the model meets the requirements of the study.

**Table 1.** Quantitative verification of tidal level, tidal current and salinity.

| Item | | Coefficient | | |
|---|---|---|---|---|
| Factor | Water Layer | CC | SS | RMSE |
| Tide level (m) | / | 0.94 | 0.92 | 0.09 |
| Current speed (m/s) | Surface layer | 0.76 | 0.77 | 0.11 |
| | Middle layer | 0.81 | 0.82 | 0.09 |
| | Bottom layer | 0.78 | 0.79 | 0.11 |
| Current direction (°) | Surface layer | 0.80 | 0.78 | 15.2 |
| | Middle layer | 0.87 | 0.83 | 11.3 |
| | Bottom layer | 0.83 | 0.82 | 12.4 |
| Salinity (‰) | Surface layer | 0.76 | 0.69 | 0.11 |
| | Middle layer | 0.72 | 0.71 | 0.11 |
| | Bottom layer | 0.78 | 0.75 | 0.12 |

*2.3. Case Study Setting*

The main driving forces behind saltwater intrusion in the Changjiang River Estuary generally involve runoff, wind, wave and tidal power. However, due to the influence of estuary morphology and wading structures, wave energy dissipates rapidly after entering the estuary [29], which makes it difficult for it to significantly affect the large-scale water movement in the Changjiang River Estuary. Therefore, this paper mainly considers the influence of runoff, wind and tidal power on saltwater intrusion.

2.3.1. Tidal Dynamic Conditions

In order to fully reflect the hydrodynamic background of the normal tidal dynamic outside the Changjiang River Estuary in the new period, and to consider the 18.6-year period of tides, the monthly tide level process with a cumulative frequency of 50% over the past 20 years at Jigujiao station (Figure 3) is selected from the hourly tide level series during the past two decades and is regarded as the simulation period for general tidal dynamic processes (Figure 4).

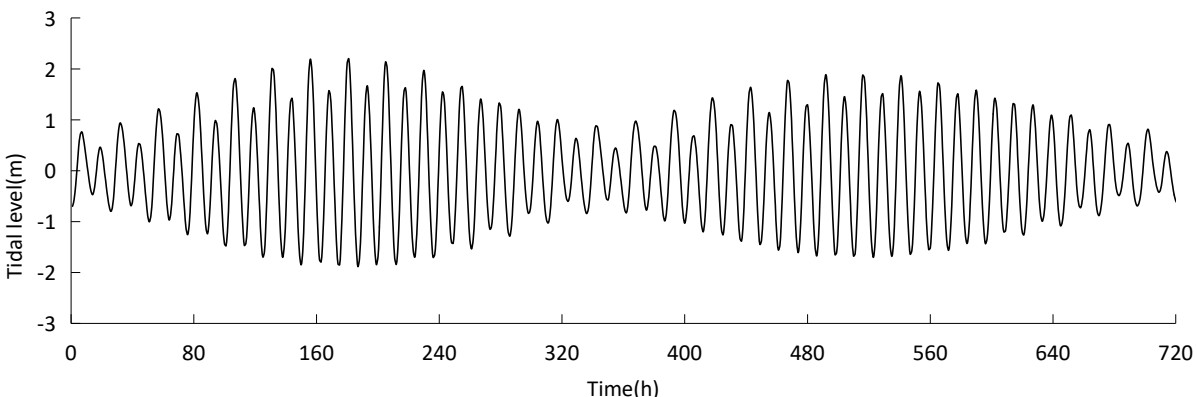

**Figure 4.** The general tide process at Jigujiao station.

2.3.2. Runoff Conditions

The riverine boundary is determined from statistical results for measured daily discharge data from the Datong Hydrological Survey Station. In order to fully consider the adverse background of saltwater intrusion in the Changjiang River Estuary, the most unfavorable runoff hydrological conditions are selected as the model runoff conditions.

The specific route design is as follows:

① According to the operation mechanism of the Three Gorges Reservoir, the annual discharge at the Datong Hydrological Survey Station will be significantly adjusted after

2003. Therefore, Datong Hydrological Survey Station daily discharge from 2003 to 2016 is selected to produce statistics that can more truly reflect the annual runoff distribution after the opening of the Three Gorges Reservoir.

② According to the design of the East Route of the South-to-North Water Transfer Project, the period of water abstraction in the year is concentrated between January and June and October and December. Therefore, here the corresponding Datong Hydrological Survey Station discharge series are adopted to generate statistics. Then, the driest monthly discharge process, with a guaranteed rate of 98% and monthly average discharge of 10,655 $m^3/s$, is obtained to be the model runoff conditions.

### 2.3.3. Water Abstraction Discharge Conditions in the East Route of the South-to-North Water Transfer Project

The model water abstraction discharge is 500 $m^3/s$ in phase I of the East Route of the South-to-North Water Transfer Project, and is 1000 $m^3/s$ in phase II.

### 2.3.4. Case Study Setting

Based on the tidal-dynamic conditions, runoff conditions, wind-field conditions and water-abstraction-discharge conditions in the East Route of the South-to-North Water Transfer Project, case studies were chosen and are shown in Table 2.

**Table 2.** Cases studies.

| Tidal-Dynamic Conditions | Runoff Conditions | Wind-Field Conditions | Water Abstraction Discharge of the East Route of the South-to-North Water Transfer Project ($m^3/s$) | | |
|---|---|---|---|---|---|
| General tidal process with a cumulative frequency of 50% | Monthly average discharge with guaranteed rate of 98% | Multi-year monthly average wind field during January and February in the dry season | CASE0 | CASE1 | CASE2 |
| | | | 0 | 500 (Phase I) | 1000 (Phase II) |

## 3. Results and Discussion

### 3.1. Influence of the East Route of the South-to-North Water Transfer Project on Estuarine Saltwater Intrusion

Due to the top rush impact of runoff and tide power, and the significant uneven distribution of runoff during the year, the dry season is often the most serious period of saltwater intrusion in the Changjiang River Estuary. Therefore, the characteristics of saltwater intrusion in the Changjiang River Estuary during dry season can be analyzed in detail.

Figure 5 shows the distribution characteristics of saltwater intrusion in different water layers during the spring flood tide and spring ebb tide in the Changjiang River Estuary against the background of differing water abstraction discharge from the East Route of the South-to-North Water Transfer Project.

As the figure shows, when the water abstraction discharge from the East Route of the South-to-North Water Transfer Project is 0 $m^3/s$ (CASE0), the distance of saltwater intrusion during the flood tide is significantly greater than that during the ebb tide. Moreover, the combined effect of low runoff and strong tide power in the dry season enhances vertical mixing in the estuary, making the saltwater intrusion characteristics in the surface, middle and bottom layers particularly similar during the tidal cycle.

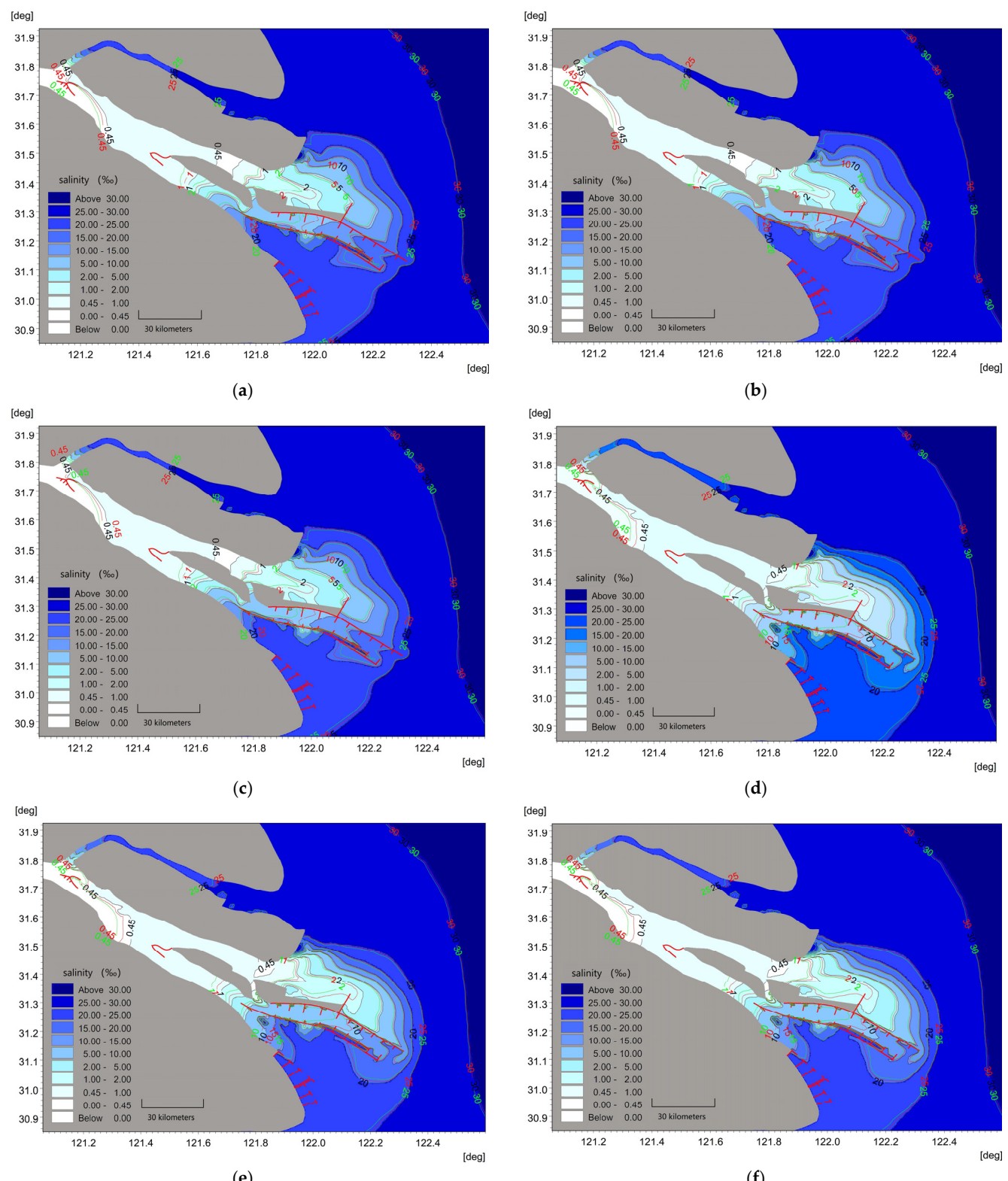

**Figure 5.** Distribution characteristics of saltwater intrusion in surface (**a**), middle (**b**) and bottom (**c**) layers of the Changjiang River Estuary during spring flood tide and in surface (**d**), middle (**e**) and bottom (**f**) layers during spring ebb tide under water abstraction from the East Route of the South-to-North Water Transfer Project. (Black line represents CASE0, red line represents CASE1, green line represents CASE2).

The saltwater intrusion is significantly weaker in the South Branch than that in the North Branch. This is because the diversion ratio in the North Branch is almost below 4% is much smaller than that of the South Branch; as a result, the effect of freshwater diluting saltwater is very weak, and the North Branch is entirely covered by high saltwater during the tidal cycle. Thus, the salinity of the different water layers below Qinglonggang are predominantly above 20‰. At the same time, the high salinity water mass from the North Branch flows backward into the South Branch during the flood tide, and is then transported downstream by the tidal current during the ebb tide, before mixing with the intruding saltwater in the North Channel and the South Channel. Therefore, the saltwater intrusion in the South Branch is dominated by the transport and diffusion of the saltwater mass coming from the North Branch.

In the North Channel and South Channel, the saltwater intrusion is mainly dominated by the confluence of saltwater from the North Branch and saltwater directly intruding from the open sea. Meanwhile, the salinity in the South Channel is higher than that in the North Channel. During the flood tide, the salinity of the different water layers in the South Channel is between 0.5‰ and 10‰, but that in the middle reach of the North Channel is below 0.45‰, and those in the upper and lower reaches of the North Channel are between 0.45‰ and 5‰. During the ebb tide, the salinity of different water layers in the South Channel is mostly between 0.5‰ and 5‰, and that in the North Channel is mostly between 0.45‰ and 2‰, except in the lower reach where the salinity is lower than 0.45‰.

In the North Passage and the South Passage, the saltwater intrusion is dominated by direct seawater intrusion, and the strength of saltwater intrusion in the South Passage is clearly higher than that in the North Passage. During the flood tide, the salinity of the different water layers in the South Passage is mostly above 15‰, and that in the North Passage is above 5‰. During the ebb tide, the downward trend in high concentration saltwater is clear, but the salinity of the different water layers in the South Passage and the North Passage still remains above 5‰.

Due to water abstraction in phase I of the East Route of the South-to-North Water Transfer Project (CASE1), the runoff into the sea decreased by about 5% on average, resulting in an increase in saltwater intrusion in each part of the estuary. However, the decrease in runoff is limited, thus the overall characteristics of saltwater intrusion in the estuary have not fundamentally changed. The distribution of salinity in the different water layers in each river section during tide cycle is particularly similar to that before the introduction of the East Route of the South-to-North Water Transfer Project. Wherein, during the flood and ebb tide, the salinity of the different water layers in the South Passage and North Passage is mostly above 5‰, and the salinity in the South Passage is greater than that in the North Passage.

Moreover, with the water abstraction in phase II of the East Route of the South-to-North Water Transfer Project (CASE2), the overall distribution characteristics of saltwater intrusion in the estuary have also not changed fundamentally, due to the limited decrease in runoff into the sea (about 9%). But the strength of the saltwater intrusion is further strengthened compared with that in CASE1.

In the case of water abstraction in the East Route of the South-to-North Water Diversion Project, the volume of discharged runoff is small, so the intensity of the salt water in the North Branch backing up into the South Branch is larger, and the average salinity of the South Branch at this time is more than 4‰; while in the case of no water abstraction, due to the discharge runoff volume being larger, the runoff dilution of the brine and the blocking effect make the degree of salt water backing up into the North Branch a little lower, and the salinity of the South Branch is not more than 2‰. With a withdrawal rate of 500 $m^3$/s, the salinity of each water layer in the North Channel and the South Channel is greater than 0.45‰, and the phenomenon that the salinity in part of the water layer of the North Channel is less than 0.45‰ in CASE0 disappears in CASE1, which is unfavorable to the withdrawal of water from Qingcaosha Reservoir. The North Branch is also still occupied by highly concentrated saline water, and the South Branch still shows that the salinity of

the different water layers near the South Bank in the middle and upper reaches is less than 0.45‰, while the salinity of other river sections is above 0.45‰. At the withdrawal rate of 1000 m$^3$/s, the impact of water transfer on saline water intrusion is further strengthened, in which the impact on the mouth of the southern trough of the Yangtze River is more clear, and the salinity will be increased by 0.42‰ to 0.47‰.

### 3.2. Intrusion Distance of 0.45‰ Iso-Salinity Front

According to the 'Standards for drinking water quality' [30], the critical salinity of drinking water is 0.45‰. Based on this, it can be seen from Figure 5 that, during the flood and ebb tide, the 0.45‰ iso-salinity front basically oscillates in the upper and middle reaches of the South Branch. Therefore, this paper takes the bifurcation of the North Channel and South Channel as the starting point, and then performs a statistical analysis on the intrusion distance of the 0.45‰ salinity front during the flood and ebb tide.

Figure 6 shows the variation in the intrusion distance of the corresponding 0.45‰ salinity front with the background of water abstraction in the East Route of the South-to-North Water Transfer Project.

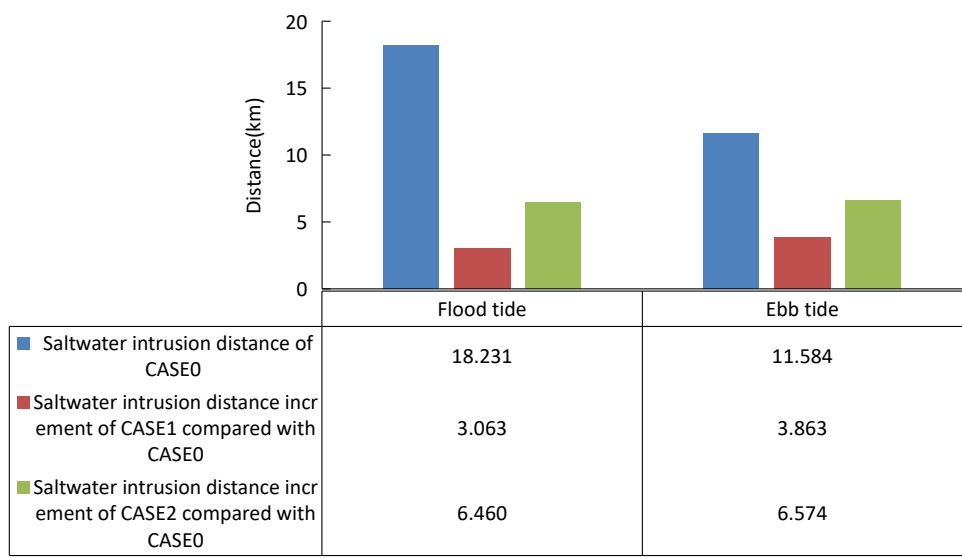

**Figure 6.** The intrusion distance of the 0.45‰ salinity front in CASE0, the increment of intrusion distance of 0.45‰ salinity front in CASE1 and CASE2 compared with that in CASE0 (km).

As the figure shows, in CASE0, the 0.45‰ iso-salinity line during the period of the flood tide and ebb tide intrudes up to about 18.231 km and 11.584 km above the bifurcation of the South Channel and North Channel.

Then, with the increase in water abstraction from 500 m$^3$/s (CASE1) to 1000 m$^3$/s (CASE2), the intrusion distance of the 0.45‰ iso-salinity line moves by about 3.063 km in CASE1 and 6.460 km in CASE2, respectively, during the flood tide, while the intrusion distance of the 0.45‰ iso-salinity line moves by about 3.863 km in CASE1 and 6.574 km in CASE2, respectively, during the ebb tide. This indicates that the water abstraction in the East Route of the South-to-North Water Transfer Project will continue to aggravate the saltwater intrusion in the estuary. Meanwhile, the intrusion distance of the 0.45‰ salinity front increases with the increase in water abstraction discharge, and the increment in the intrusion distance in CASE2 is about 1.9 times higher than that in CASE1, which is similar to the increment in water abstraction discharge (2 times).

However, considering that the reach of about 220 km from the tidal current limit to the mouth bar of the Changjiang River Estuary is the main mixture area for fresh and salt water [31], the increment in the intrusion distance of the 0.45‰ iso-salinity line caused by water abstraction in phase I (CASE1) and phase II (CASE2) of the East Route of the

South-to-North Water Transfer Project accounts for only 1.8% and 3.0% of the length of this river reach, respectively. Moreover, they are also significantly smaller than the percentages of 4.7% and 9.4% for the water abstraction discharge which account the proportion of the driest monthly average discharge with a guaranteed rate of 98%. This also shows that the change in saltwater intrusion caused by the water abstraction in the East Route of the South-to-North Water Transfer Project is actually limited.

### 3.3. Estimation of Freshwater Resources in the Estuary

Similarly, according to the critical salinity of 0.45‰ (GB 5749-2022) [30], the changes in freshwater resources in the Changjiang River Estuary against background of water abstraction in the East Route of the South-to-North Water Transfer Project are statistically analyzed.

Affected by the periodic variation of tidal dynamics, the total amount of freshwater resources in the estuary also changes periodically during the tide cycle. Therefore, this paper uses the volume of freshwater with a salinity below 0.45‰ during the flood and ebb tides to estimate the change in freshwater resources.

Figure 7 shows the statistical results for the volume of freshwater with a salinity below 0.45‰ in the estuary below the Xuliujin during the flood and ebb tides under background of water abstraction of the East Route of the South-to-North Water Transfer Project.

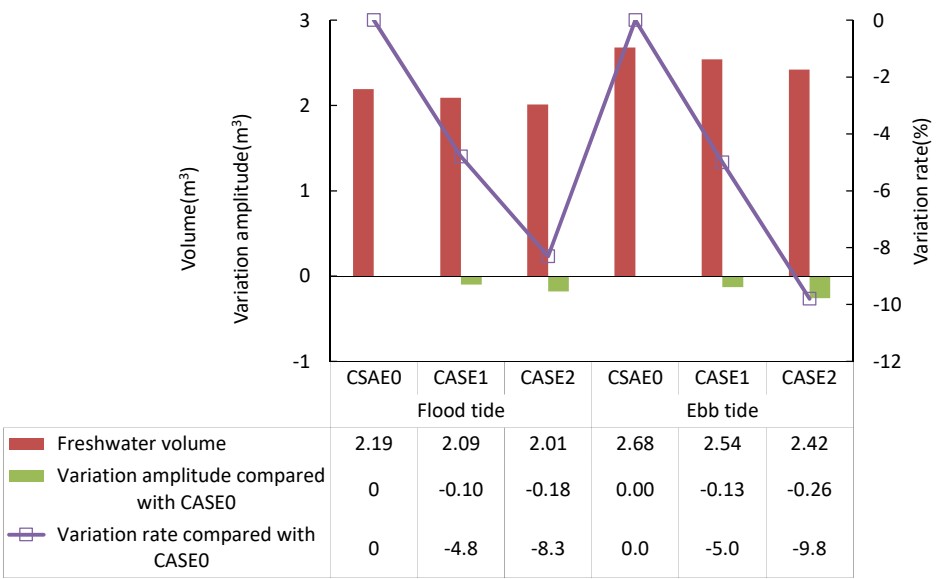

**Figure 7.** Statistics for freshwater resources with salinity less than 0.45‰ in estuary during flood and ebb tide against the background of water abstraction in the East Route of the South-to-North Water Transfer Project.

It can be seen that, with the increase in water abstraction discharge from 0 $m^3$/s (CASE0) to 500 $m^3$/s (CASE1) and 1000 $m^3$/s (CASE2), the volume of freshwater with a salinity below 0.45‰ in the estuary decreases from 2.19 $km^3$ to 2.09 $km^3$ and 2.01 $km^3$ during the flood tide, with variation rates of −4.8% and −8.3%. During the ebb tide, with the increase in water abstraction discharge, the volume of freshwater with a salinity below 0.45‰ in the estuary decreases from 2.68 $km^3$ to 2.54 $km^3$ and 2.42 $km^3$, with a variability of −5.0% and −9.8%. Thus, it can be deemed that, against the background of the East Route of the South-to-North Water Transfer Project, the volume of freshwater with a salinity below 0.45‰ during the flood tide is always smaller than that during the ebb tide, and the fresh water volume shows a clear trend of decreasing with the increase in water abstraction. It has been shown that the reduction in estuarine freshwater resources caused by the East Route of the South-to-North Water Transfer Project is still limited, and the variation rate is mostly within −10%.

At the same time, due to the high saltwater content of the North Branch flowing backward into the upper reach of the South Branch and spreading to the North Channel and South Channel, the salinity profile between the South Branch and the North Channel (South Channel) appears saddle shape (Figure 5), and the salinity of the different water layers near the north bank is higher than that near the south bank in the South Branch. This makes the freshwater resources in the upper and middle reaches of the South Branch mainly concentrated near the southern bank, which is not conducive to the protection of a suitable water intake time for the Dongfengxisha Reservoir (DFXSR) on the north bank of the South Branch, but is beneficial for the Baogang Reservoir (BGR) on the south bank of the South Branch.

### 3.4. Unsuitable Water Intake Times for Reservoirs in the Estuary

There are many reservoirs for drinking water in Shanghai, wherein the top three are located along the main stream of the Changjiang River Estuary, namely Qingcaosha Reservoir (QCSR), BGR and DFXSR (Figure 3). Their total storage capacity accounts for more than 90% of the total storage capacity of all of the reservoirs in Shanghai. Saltwater intrusion is characterized by the superimposition of a positive intrusion of seawater from the outer sea and the backflow of seawater from the North Branch, with positive intrusion being the main feature which has continuously exceeded the standard of chloride content at the water intake and seriously affected the safety and quality of the water [32].

According to the saltwater intrusion standard of the Shanghai Water Supply Company, the critical index of salinity is 0.18‰. Therefore, based on this standard, the unsuitable water intake times for these three reservoirs can be determined.

Figure 8 shows the statistical results of the unsuitable water intake time of each reservoir within one month and the increment in unsuitable water intake days compared with CASE0.

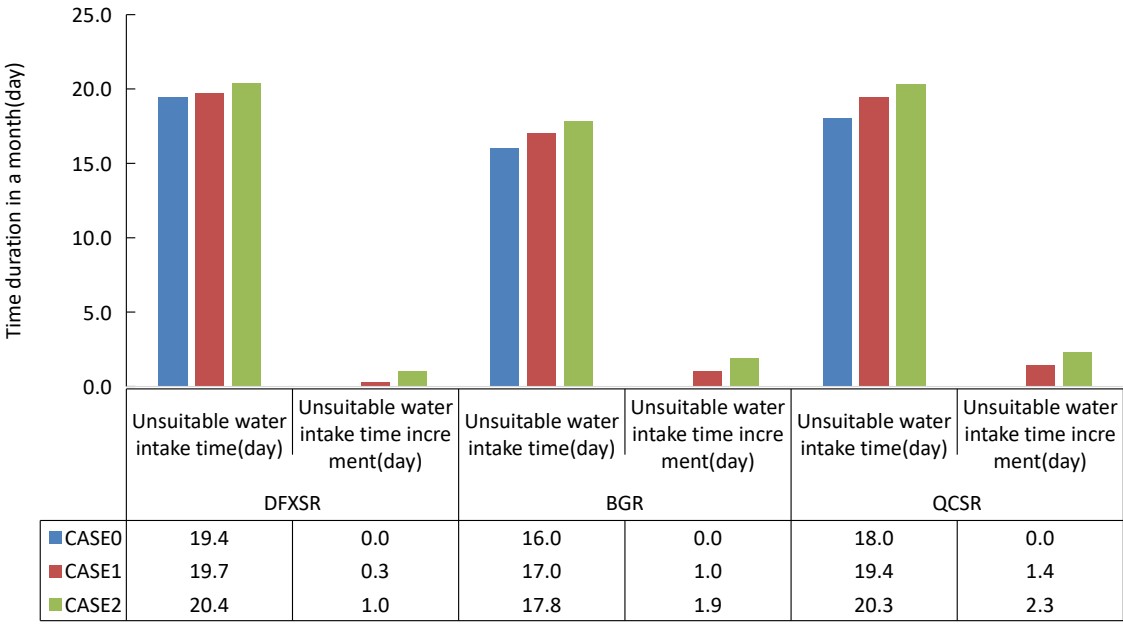

**Figure 8.** Unsuitable water intake days in a month of reservoirs and increments compared with CASE0.

As the figure shows, when the water abstraction discharge increases from 0 m$^3$/s (CASE0) to 500 m$^3$/s (CASE1) and 1000 m$^3$/s (CASE2), the value of unsuitable water intake days in a month for DFXSR increases from 19.4 days to 20.4 days. And the value for BGR increases from 16.0 days to 17.8 days, meanwhile of the value for QCSR increases from 18.0 days to 20.3 days.

Therefore, it can be found that, with a continuous increase in water abstraction from the East Route of the South-to-North Water Transfer Project, the number of unsuitable

water intake days within a month at different reservoirs shows a gradual increase trend. Among them, compared with CASE0, the maximum increment in unsuitable water intake days for DFXSR is about 1.0 days, and for BGR it is about 1.9 days; meanwhile, for QCSR it is about 2.3 days. However, from the perspective of the range of the change in unsuitable water intake time, the impact of the East Route of the South-to-North Water Transfer Project is still limited, and the increment in unsuitable water intake time is basically within 13%.

Furthermore, it should be noted that the unsuitable water intake days for QCSR in the lower reach of the South Branch and DFXSR in the upper reach of the South Branch are relatively larger than for BGR in the middle reach of the South Branch. Based on our analysis, this is mainly due to the high saltwater of the North Branch flowing backward into the South Branch and being transported downward along the north bank of the South Branch and then mixing with the saltwater in the North Channel (Figure 5).

*3.5. Ecological Discharge of Estuarine Saltwater Intrusion in Dry Season*

The Changjiang River Estuary has a large runoff and a strong water self-purification capacity. Since the runoff directly affects the frequency and intensity of saltwater intrusion, it also directly affects the estuarine freshwater resources and water intake safety of reservoirs [29,31,33,34]. Therefore, here, the ecological discharge of the Changjiang River Estuary is defined as the riverine discharge required to maintain the appropriate salinity of the estuary.

In order to reflect the changes in saltwater intrusion in the estuary with different runoffs, and combined with the actual water abstraction that may occur for the East Route of the South-to-North Water Transfer Project, more simulated cases are re-set here. The details are shown in Tables 3 and 4.

**Table 3.** New detailed cases considering water abstraction of the East Route of the South-to-North Water Transfer Project.

| Tidal Dynamic Conditions | Wind Field Conditions | Runoff Conditions | Water Abstraction Discharge in the East Route of the South-to-North Water Transfer Project (m$^3$/s) | | | | | | |
|---|---|---|---|---|---|---|---|---|---|
| General tidal process with a cumulative frequency of 50% | Multi-year monthly average wind field during January and February in the dry season | Monthly average discharge with 98% guarantee rate | CASE0 | CASE1 | CASE2 | CASE3~CASE7 | CASE8 | CASE9 | CASE10 |
| | | | 0 | 100 | 200 | 300~700 | 800 | 900 | 1000 |

**Table 4.** New detailed cases considering typical Datong Hydrological Survey Station discharge.

| Tidal Dynamic Conditions | Wind Field Conditions | Water Abstraction in the East Route of the South-to-North Water Transfer Project (m$^3$/s) | Runoff Conditions (m$^3$/s) | | | |
|---|---|---|---|---|---|---|
| | | | CASE11 | CASE12 | CASE13 | CASE14 |
| General tidal process with a cumulative frequency of 50% | Multi-year monthly average wind field during January and February in the dry season | 0 | 6730 (Historical monthly minimum discharge during 1950~2016) | 8000 | 20,000 | 22,610 (Monthly average discharge from October to next June during 2003~2016) |

Meanwhile, considering that monthly variation in runoff clearly always exists throughout a year, as a result, the saltwater intrusion becomes similar in the Changjiang River Estuary. Thus, the monthly average variation characteristics are used to comprehensively reflect the relationship between saltwater intrusion and ecological discharge in a relatively typical duration.

Figure 9 shows the relationship between the monthly average river discharge into estuary and the monthly average salinity in the bifurcation of the South Branch and North Branch. It should be noted that, here, the river discharge into the estuary is defined as being a combination of the Datong Hydrological Survey Station discharge minus the water abstraction discharge from the East Route of the South-to-North Water Transfer Project, which represents the real amount of riverine runoff entering into the Changjiang River Estuary and acting on saltwater intrusion.

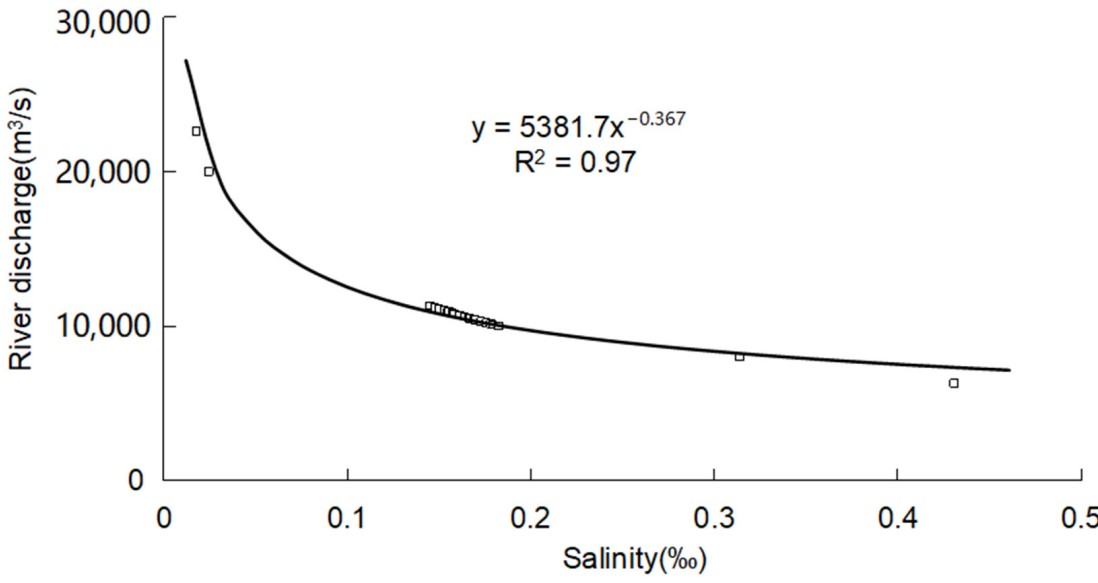

**Figure 9.** The relationship between the monthly average riverine discharge into estuary and the monthly average salinity of the bifurcation of the North Branch and South Branch.

As the figure shows, the relationship between the monthly average salinity and the monthly average riverine discharge into the estuary presents a clear power function relationship. Thus, the power function can be used to create the corresponding relationship between salinity and discharge. The result is as follows.

$$Q = 5381.7S^{-0.367} \tag{4}$$

$$R^2 = 0.97 \tag{5}$$

where, Q is the monthly average riverine discharge into estuary, m$^3$/s; S is the monthly average salinity in the bifurcation of the North Branch and South Branch, ‰.

In order to meet the estuarine ecological water requirements, the saltwater intrusion standard of Shanghai Water Supply Company is selected as the control salinity to estimate the critical ecological discharge of the estuary. Then, combined with Figure 9 and Equation (1), the ecological discharge corresponding to a saltwater intrusion standard of 0.18‰ is proposed to be 10,098 m$^3$/s. Q and Datong hydrological gauging station runoff minus the South-to-North Water Diversion East Line water intake runoff obtained by the difference (q); for comparison, if Q ≤ q, then the South-to-North Water Diversion East Line water intake work can be carried out normally; if Q > q, then the South-to-North Water Diversion East Line intake work should be reduced or stopped; you can appropriately increase the upper reaches of the water conservancy projects such as the discharge of the flow rate, and

change the scheduling of the downstream estuarine area to mitigate the impacts of salt water intrusion.

According to the history of saltwater intrusion in the Changjiang River Estuary, the saltwater intrusion during February 2014 was the most persistent saltwater intrusion that almost caused a major ecological disaster in the Changjiang River Estuary [22]. To alleviate the impact of the salty tide intrusion, the Three Gorges Reservoir has stepped up its water replenishment efforts, increasing the discharge flow by 1000 m$^3$/s. (Shanghai Yangtze River mouth water source suffered the longest salty tide invasion (www.gov.cn (accessed on 6 January 2024)). Compared with the monthly average discharge at the Datong Hydrological Survey Station of 11,300 m$^3$/s, the ecological discharge proposed by Equation (1) is slightly smaller. From our analysis, it seems that this is because there are more than 200 sluices and pumping stations on both sides of the Changjiang River below the Datong Hydrological Survey Station, and their designed water abstraction capacity has far exceeded the water abstraction discharge of the East Route of the South-to-North Water Transfer Project [35]. Therefore, during February 2014, in order to meet the needs of local industrial and agricultural production and residents' life, the sluices and pumping stations on both sides of the river inevitably continued to abstract water from the river. Thus, the effective runoff that really entered into the Changjiang River Estuary and played a real role in suppressing saltwater intrusion was actually less than the relevant monthly average discharge at the Datong Hydrological Survey Station.

Thus, it can be deemed, to a certain extent, that the ecological discharge proposed here is reasonable.

## 4. Conclusions

This paper uses a three-dimensional hydrodynamic turbulence model to establish a saltwater intrusion numerical model for the Changjiang River Estuary. On the basis of similar model verifications, for the first time, the built model was used to study the difference among the East Route of the South-to-North Water Transfer Project and runoff volume saltwater intrusion distance and water layer during the new period of the Three Gorges Reservoir operation after 2003. Regarding the influence of salinity, the main research conclusions are:

The dry season in the Changjiang Estuary sees severe saltwater intrusion due to low runoff and tidal forces. In the East Route (CASE0), flood tide intrusion surpasses that seen during the ebb tide. Vertical mixing equalizes intrusion across layers. Intrusion in the North Branch exceeds that in the South Branch due to lower diversion. In the North and South Channels, intrusion is influenced by North Branch saltwater and the open sea. CASE1 slightly increases intrusion; CASE2 intensifies intrusion with a limited decrease in runoff.

Analyzing water abstraction's impact (CASE0 to CASE2), flood tide intrusion increases from 18.231 km to 24.691 km, and ebb tide distances increase from 11.584 km to 18.058 km. Water abstraction intensifies saltwater intrusion, with CASE2 showing a 1.9 times higher increment than CASE1. However, the intrusion increase is limited, accounting for only 1.8% (CASE1) and 3.0% (CASE2) of the 220 km main mixture area in the estuary.

Combined with the actual situation, the simulated saltwater intrusion was again simulated, and then the relationship between the monthly average riverine discharge into estuary and monthly average salinity of the Changjiang River Estuary was also revealed. Then, the critical ecological discharge that controls the salinity and ecology of the Changjiang River Estuary is produced at the end. In contrast, freshwater intrusion has had negative impact on primary and secondary plankton and benthic organisms in the sea, but the mechanisms by which freshwater intrusion is affected by various elements in estuaries are more similar to those of saltwater intrusion. Therefore, the numerical model in this paper can be used to study the intrinsic mechanisms of freshwater intrusion in the corresponding estuaries.

Against the background of the estuarine environment and coastal ecology protection and restoration, the conclusions of this paper have important implications for the research on the operation of integrated multi-projects for suppressing saltwater intrusion in the new period following the operation of the Three Gorges Reservoir after 2003.

**Author Contributions:** Conceptualization, H.H., Y.W. and S.W.; numerical model, H.H. and Y.W.; software, H.H.; validation, H.H., Y.W. and X.H.; formal analysis, H.H.; data curation, Y.W.; writing— original draft preparation, H.H.; writing—review and editing, H.H., Y.W. and S.W.; supervision, S.W., Y.L. and X.H.; project funding acquisition, Y.W. All authors have read and agreed to the published version of the manuscript.

**Funding:** This research was funded by the National Natural Science Foundation of China (NO. 51979096); the National Natural Science Foundation of China (NO. U2040203); National Key R&D Program of China (NO. 2022YFC3106103); Fujian Provincial Key Laboratory of Coast and Island Management Technology Study (NO. FJCIMTS2022-03).

**Institutional Review Board Statement:** Not applicable.

**Informed Consent Statement:** Not applicable.

**Data Availability Statement:** Data are contained within the article. The data presented in this study are available in Tables 1–4 and Figures 2, 4 and 6–9.

**Acknowledgments:** Joint Fund of the Ministry of Education for Equipment Pre research (NO. 8091B022123); the Key Laboratory of Coastal Disasters and Defense, Ministry of Education, Hohai University Program (The influence of the Deepwater Channel Regulation Project on salt The influence of the Deepwater Channel Regulation Project on saltwater intrusion in the Yangtze River Estuary); the Fundamental Research Funds for the Central Public Welfare Research Institutes, Nanjing Hydraulic Research Institute Welfare Research Institutes, Nanjing Hydraulic Research Institute (NO. YN912001).

**Conflicts of Interest:** The authors declare no conflicts of interest.

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
