# Peer review of "Saltwater Intrusion in the Changjiang River Estuary in Response to the East Route of the South-to-North Water Transfer Project in the New Period after 2003"

_sustainability, doi:10.3390/su16020683_

Round 1
Reviewer 1 Report
Comments and Suggestions for Authors
I have read the paper with great attention and interest. It seems to me to be a good approach to the problem of interference between the sea and the water table of the rivers in that area. Perhaps some reference could have been made to the fact that the rise in sea level due to interglacial is making the aquifers in the area progressively more vulnerable. I consider that the approach to the problem is perfect and that it has been developed in a balanced way.
The same model of addressing the problem can be applied to other similar cases, essentially coasts such as those of the Northwest of Spain where the river network is totally intervened by hydroelectric exploitation, which causes the opposite phenomenon to that described here: The intrusion of fresh water into the coastal environment with interesting effects of desalination of shellfish farming areas with ecological catastrophes during large avalanches of fresh water to the sea.
Author Response
Author's Reply to the Review Report (Reviewer 1)
Dear reviewer,
Re: Manuscript ID: sustainability-2750369 and Title: Saltwater intrusion of the Changjiang River Estuary responding to the East Route of South to North Water Transfer Project in the new period after 2003.
Thank you for your letter and the reviewers’ comments concerning our manuscript entitled “Saltwater intrusion of the Changjiang River Estuary responding to the East Route of South to North Water Transfer Project in the new period after 2003” (sustainability-2750369). Those comments are valuable and very helpful. We have read through comments carefully and have made corrections. Based on the instructions provided in your letter, we uploaded the file of the revised manuscript. Revisions in the text are shown using red highlight for additions, and strikethrough font for deletions. The responses to the reviewer's comments are marked in red and presented following.
We would love to thank you for allowing us to resubmit a revised copy of the manuscript and we highly appreciate your time and consideration.
Sincerely,
Yan Wang.
Reviewer 1:
Q1: I have read the paper with great attention and interest. It seems to me to be a good approach to the problem of interference between the sea and the water table of the rivers in that area. Perhaps some reference could have been made to the fact that the rise in sea level due to interglacial is making the aquifers in the area progressively more vulnerable. I consider that the approach to the problem is perfect and that it has been developed in a balanced way.
The same model of addressing the problem can be applied to other similar cases, essentially coasts such as those of the Northwest of Spain where the river network is totally intervened by hydroelectric exploitation, which causes the opposite phenomenon to that described here: The intrusion of fresh water into the coastal environment with interesting effects of desalination of shellfish farming areas with ecological catastrophes during large avalanches of fresh water to the sea.
Response: Thank you for your confirmation of the article, this paper on the study of saltwater intrusion in the Changjiang River Estuary, focusing on the East Route of the South-to-North Water Transfer Project various water withdrawals and runoff volume of the Changjiang River Estuary on the saltwater intrusion, in addition to this, the rise in sea level will also have an impact on the saltwater intrusion, and it is a long-term consequence of The impact of sea level rise on saltwater intrusion is also expected to be a long-term consequence of sea level rise. Sea level rise will increase seawater inundation and promote salt water intrusion, increasing intrusion distance and surface water salinity, which may lead to a more fragile and graded aquifer salinity and a continuous exceedance of chloride content. In this paper we have also fitted the relationship between monthly mean salinity and monthly mean discharge to further monitor saltwater intrusion into the estuary.
In addition, unlike saltwater intrusion, freshwater intrusion can also negatively affect marine plankton or benthic organisms, although the mechanisms by which freshwater intrusion is affected by various elements in the estuary are more similar to those of saltwater intrusion. Therefore, the methodology of this paper can be used to study the intrinsic mechanisms of freshwater intrusion in the corresponding estuaries.
Corresponding content has been added to the Lines 63, 64 and 519 to 524 of the article.

Reviewer 2 Report
Comments and Suggestions for Authors
Paper raises a pertinent concern about the relationship between saltwater intrusion and runoff in the Changjiang River Estuary (CRE), particularly focusing on the impact of the East Route of the South-to-North Water Transfer Project (ERSNWTP) on saltwater intrusion patterns.
1. Could you elaborate on the numerical simulation methodology employed to analyze the impact of ERSNWTP on saltwater intrusion in CRE?
2. Detailed information on model inputs, equations used, and validation strategies would enhance the study's credibility and replicability.
3. How comprehensive was the dataset used to model the relationship between ERSNWTP water abstraction and saltwater intrusion patterns? Were various scenarios and sensitivity analyses conducted to gauge potential fluctuations?
4. How was the hydrodynamic model integrated into the numerical simulations? Exploring the nuances of model integration and its influence on predicting saltwater intrusion in response to ERSNWTP water abstraction could enhance the study's precision.
5. Did the study account for the temporal resolution of data in assessing saltwater intrusion patterns? Analyzing seasonal or monthly variations in saltwater intrusion due to changing runoff patterns might reveal nuanced impacts.
6. Beyond salinity changes, did the study delve into the potential implications of altered saltwater intrusion on the water quality in the estuary?
7. Was a sensitivity analysis conducted to examine the robustness of the model outcomes concerning various parameters, such as tidal influence, river discharge variability, or climate change scenarios?
Comments on the Quality of English LanguageModerate editing of English language required
Author Response
Author's Reply to the Review Report (Reviewer 2)
Dear reviewer,
Re: Manuscript ID: sustainability-2750369 and Title: Saltwater intrusion of the Changjiang River Estuary responding to the East Route of South to North Water Transfer Project in the new period after 2003.
Thank you for your letter and the reviewers’ comments concerning our manuscript entitled “Saltwater intrusion of the Changjiang River Estuary responding to the East Route of South to North Water Transfer Project in the new period after 2003” (sustainability-2750369). Those comments are valuable and very helpful. We have read through comments carefully and have made corrections. Based on the instructions provided in your letter, we uploaded the file of the revised manuscript. Revisions in the text are shown using red highlight for additions, and strikethrough font for deletions. The responses to the reviewer's comments are marked in red and presented following.
We would love to thank you for allowing us to resubmit a revised copy of the manuscript and we highly appreciate your time and consideration.
Sincerely,
Yan Wang.
Reviewer 2:
Q1: Could you elaborate on the numerical simulation methodology employed to analyze the impact of ERSNWTP on saltwater intrusion in CRE?
Response: We are grateful for the suggestion make the following changes to clarify. The numerical simulation methodology employed in this paper: Based on the measured hydrological salinity data in the Changjiang River Estuary during the dry season, a three-dimensional mathematical salinity model was constructed using the MIKE21Flow Model FM unstructured grid hydrodynamics (HD) module developed by the Danish company DHI and its associated thermo-saline transport (TS) module. Based on the validation of the mathematical model, the spatial and temporal variations of flow and salinity in the Changjiang River Estuary during the dry season are analysed in detail and the backwater salt fluxes of the northern saline water are preliminarily calculated.
Relevant content is added to lines 122 to 135 of the article.
Q2: Detailed information on model inputs, equations used, and validation strategies would enhance the study's credibility and replicability.
Response: We are grateful for the suggestion make the following changes to clarify.
Model inputs: The tidal and runoff conditions were selected based on an analysis of previous hydrographic data. The topographic input conditions were obtained from underwater topographic data from 2002 to 2017. The model input wind field conditions were derived from multi-year monthly mean wind field data in January and February during the dry season. The three offshore boundaries were driven by a global tidal model (Matsumoto et al., 2000). The river boundaries are determined by measured flow data from the Datong hydrological station (Relevant content is added to lines 136 to 151 of the article).
Equations used: The equations used in this paper have been fully applied and validated in previous studies on saltwater intrusion modelling and have been acknowledged by many scholars (Chen, et al., 2018; Lin, et al., 2019; Yang, et al., 2013) (Relevant content is added to lines 172 to 174 of the article.).
Validation strategies: The data used for model validation are measured data, including tide level, flow velocity and direction, and salinity throughout the spring, middle and neap tide period. CC (correlation coefficient) and SS (skill score) were used for the comprehensive evaluation of the model calculation data, with larger values indicating that the model simulation results were about good, where SS was evaluated according to Allen's classification method (Allen, et al., 2007). Relevant content is added to lines 139 to 142 and 191 to 194 of the article.
Q3: How comprehensive was the dataset used to model the relationship between ERSNWTP water abstraction and saltwater intrusion patterns? Were various scenarios and sensitivity analyses conducted to gauge potential fluctuations?
Response: We are grateful for the suggestion make the following changes to clarify.
The model's western riparian boundary is situated at Datong Hydrological Survey Station, while its three offshore boundaries are determined by a global tidal model (Matsumoto et al., 2000). These are synchronous measurements of full-tide current salinity data in the waters of the Yangtze River estuary. Allen's classification method was used to validate the model, which produced the following results: all tidal level validation correlations were greater than 0.9, and the validation correlations for current velocity and direction were also in the range of 0.76-0.87, indicating excellent simulation results. In the selection of time series, it is important to include the shortest time series that contains consecutive large, medium, and small tides. This will enable the study of the effect of saltwater intrusion (Relevant content is added to lines 136 to 138 and 157 to 161 and 203 to 208 of the article).
The article presents a sensitivity analysis of the Coe's force model and vertical stratification. The calculations were compared using four error metrics: RMSE (Root Mean Square Error), MAE (Mean Absolute Error), ME (Maximum Error), and A.D (Relative Deviation). The results indicate that the calculations worked best when Coe's force and stratification of 11 layers were present. The article does not reflect the sensitivity analysis as its focus is on exploring the effect of salinity intrusion, and the cut length is significant.
Relevant content is added to lines 167 to 169 of the article.
Q4: How was the hydrodynamic model integrated into the numerical simulations? Exploring the nuances of model integration and its influence on predicting saltwater intrusion in response to ERSNWTP water abstraction could enhance the study's precision.
Response: We are grateful for the suggestion make the following changes to clarify.
During salt water intrusion, water density increases as salinity increases. Previous studies have solely focused on the unilateral saltwater intrusion process. However, this paper proposes a coupled approach to simulate both hydrodynamics and saltwater intrusion. The simulation process takes into account the changes in water density caused by saltwater intrusion. Additionally, the water flow eddy viscosity coefficient is determined using the k-ε model, and the salinity diffusion factor is directly proportional to the water flow eddy viscosity coefficient (with a ratio of 1). This simulation method is more accurate than using uncoupled methods to simulate hydrodynamics and saltwater intrusion, and it can more accurately reflect the actual process of mixing saltwater and freshwater.
Relevant content is added to lines 131 to 135 and 172 to 177 of the article.
Q5: Did the study account for the temporal resolution of data in assessing saltwater intrusion patterns? Analyzing seasonal or monthly variations in saltwater intrusion due to changing runoff patterns might reveal nuanced impacts.
Response: We are grateful for the suggestion make the following changes to clarify.
The article examines saltwater intrusion during the dry season, which is the most severe period of saltwater intrusion in the central region due to the overhead effects of runoff and tidal forces, as well as the significant heterogeneity in the intra-annual distribution of runoff. In contrast, during the flood season, when runoff reaches 50,000 to 80,000 cubic feet per second, the impact of saltwater intrusion is minimal. The article uses data with time scales above one consecutive spring, middle and neap tide, which is sufficient to cover the different time scales of saltwater intrusion in the estuary (Relevant content is added to lines 80 to 81 and 139 to 142 of the article).
Seasonal variations affect the degree of saltwater intrusion, with the Changjiang River runoff being a decisive factor in the Changjiang River Estuary. This determines the significant difference in saltwater intrusion during the flood and dry seasons. Using the flood season in July 2010 and the dry season in January 2011 as an example, the salinity of each station remained around 0.1‰ during the flood season. However, saline intrusion still occurred to varying degrees at all stations during the dry season, with the maximum being 150 times the salinity during the flood season. The monthly mean flow did not differ much between the dry and flood seasons and had the same effect on saltwater intrusion.
Relevant content is added to lines 100 to 103 of the article.
Q6: Beyond salinity changes, did the study delve into the potential implications of altered saltwater intrusion on the water quality in the estuary?
Response: We are grateful for the suggestion make the following changes to clarify.
This paper summarizes the analysis of saltwater intrusion distance, salinity, and unsuitable times for water withdrawal based on the model's results. Specifically, it examines the distance of salt peak intrusion, estimates freshwater resources, and identifies unsuitable times for water withdrawal from reservoirs. Saltwater intrusion can increase the salinity in the estuary, which can affect the safety of water quality and prevent it from meeting domestic water standards. Additionally, literature suggests that saltwater intrusion can alter the water quality in the estuary. Saltwater intrusion is defined as the intrusion of seawater from the outer sea and the inversion of seawater from the northern branch, resulting in an increase in the chloride content of the Yangtze River estuary. This phenomenon poses a serious threat to the safety of water quality.
Relevant content is added to lines 406 to 410 of the article.
Q7: Was a sensitivity analysis conducted to examine the robustness of the model outcomes concerning various parameters, such as tidal influence, river discharge variability, or climate change scenarios?
Response: We are grateful for the suggestion make the following changes to clarify.
The article presents a sensitivity analysis of the Koch force model and vertical stratification. The results indicate that the model performs optimally when the Koch force is present and in 11 layers.
The article conducted simulation analyses on ten different water flow rates (ranging from 0m3/s to 1000m3/s) and four runoff conditions (6730m3/s, 8000m3/s, 20000m3/s, 22610m3/s). The model calculations were stable and the results were fitted to the salinity-flow correspondence using a power function.
Relevant content is added to lines 167 to 169 and 466 to 471 of the article.

Reviewer 3 Report
Comments and Suggestions for Authors
The topic of the article is interesting but:
The authors use a large number of abbreviations, making the text difficult to read,
the phenomenon of water entering river estuaries is common in the world, it is not described here and no similar examples are provided in the literature
the drawings are of poor quality:
there is no linear scale,
we don't mark maps like that on maps,
in Figure 5 we have different colors of overlapping fonts, it is not legible,
there is an unlabeled legend in figure 5,
please mark the name of Datong and provide the commission, because the word Datong immediately brings to mind Datong, Shanxi
changes of 0.45‰ iso-salinity, please mark on one figure and differentiate, e.g. by color in individual stages, currently presented is not very legible
maybe it's worth marking TGR on one of the maps
Author Response
Author's Reply to the Review Report (Reviewer 3)
Dear reviewer,
Re: Manuscript ID: sustainability-2750369 and Title: Saltwater intrusion of the Changjiang River Estuary responding to the East Route of South to North Water Transfer Project in the new period after 2003.
Thank you for your letter and the reviewers’ comments concerning our manuscript entitled “Saltwater intrusion of the Changjiang River Estuary responding to the East Route of South to North Water Transfer Project in the new period after 2003” (sustainability-2750369). Those comments are valuable and very helpful. We have read through comments carefully and have made corrections. Based on the instructions provided in your letter, we uploaded the file of the revised manuscript. Revisions in the text are shown using red highlight for additions, and strikethrough font for deletions. The responses to the reviewer's comments are marked in red and presented following.
We would love to thank you for allowing us to resubmit a revised copy of the manuscript and we highly appreciate your time and consideration.
Sincerely,
Yan Wang.
Reviewer 3:
Q1: The topic of the article is interesting but: The authors use a large number of abbreviations, making the text difficult to read.
Response: We are grateful for the suggestion make the following changes to clarify.
As suggested by the reviewer, we have changed the abbreviation in the article to the full name.
Q2: The phenomenon of water entering river estuaries is common in the world, it is not described here and no similar examples are provided in the literature.
Response: We are grateful for the suggestion make the following changes to clarify.
As suggested by the reviewer, we have added the phenomenon and principles of saltwater intrusion in lines 32 to 35 of the article.
Q3: The drawings are of poor quality: there is no linear scale.
Response: We are grateful for the suggestion make the following changes to clarify.
As suggested by the reviewer, we have modified Figure 5 to indicate the linear scale used by the article in performing the simulations.
Q4: In Figure 5 we have different colors of overlapping fonts, it is not legible.
Response: We are grateful for the suggestion make the following changes to clarify.
As suggested by the reviewer, we made changes to Figure 5 to separate the fonts in the overlapping parts of the figure to make the image more aesthetically pleasing
Q5: There is an unlabeled legend in figure 5.
Response: We are grateful for the suggestion make the following changes to clarify.
As suggested by the reviewer, we have made a modification to Figure 5 to add a note about the unlabeled legend in the figure.
Q6: Please mark the name of Datong and provide the commission, because the word Datong immediately brings to mind Datong, Shanxi.
Response: We are grateful for the suggestion make the following changes to clarify.
The 'Datong' in the article is Datong Hydrological Station, not Datong, Shanxi Province. As suggested by the reviewer, to avoid ambiguity, we have rewritten 'Datong' as Datong Hydrological Station.
Q7: Changes of 0.45‰ iso-salinity, please mark on one figure and differentiate, e.g. by color in individual stages, currently presented is not very legible.
Response: We are grateful for the suggestion make the following changes to clarify.
As suggested by the reviewer, we have made changes to Figure 5 to make the colour contrast more pronounced.
Q8: Changes of 0.45‰ iso-salinity, please mark on one figure and differentiate, e.g. by color in individual stages, currently presented is not very legible.
Response: We are grateful for the suggestion make the following changes to clarify.
As suggested by the reviewer, we marking Three Gorges Reservoir in Figure 1.

Round 2
Reviewer 2 Report
Comments and Suggestions for Authors
This research highlights the significant shifts in runoff distribution into CRE following the continuous operation of the Three Gorges Reservoir (TGR) post-2003, leading to marked alterations in the saltwater intrusion pattern within CRE. Furthermore, the proposal and advancement of the East Route of the South-to-North Water Transfer Project (ERSNWTP) are poised to induce another change in annual runoff distribution into CRE, potentially altering the saltwater intrusion pattern once more.
The study underscores the urgency of understanding how the water abstraction of ERSNWTP, particularly post-2003, affects saltwater intrusion in CRE. Employing numerical simulation methods, this paper investigates this impact, assessing saltwater intrusion distances, freshwater resources, and unsuitable timeframes for water intake by reservoirs. The findings notably highlight the considerable influence of ERSNWTP on saltwater intrusion within the CRE.
However, several academic inquiries arise from this study:
1.What specific modeling techniques were employed in the numerical simulation to assess the impact of ERSNWTP on saltwater intrusion in CRE?
2.Could this research provide a comparative analysis of saltwater intrusion patterns before and after the ERSNWTP's water abstraction?
3.How do the proposed ecological discharges intend to control saltwater intrusion, and what empirical evidence or simulations support their effectiveness?
4.Further insights into modeling approaches and empirical validations would enhance the study's depth and credibility.
Comments on the Quality of English LanguageModerate editing of English language required.
Author Response
Author's Reply to the Review Report (Reviewer 2)
Dear reviewer,
Re: Manuscript ID: sustainability-2750369 and Title: Saltwater intrusion of the Changjiang River Estuary responding to the East Route of South to North Water Transfer Project in the new period after 2003.
Thank you for your letter and the reviewers’ comments concerning our manuscript entitled “Saltwater intrusion of the Changjiang River Estuary responding to the East Route of South to North Water Transfer Project in the new period after 2003” (sustainability-2750369). Those comments are valuable and very helpful. We have read through comments carefully and have made corrections. Based on the instructions provided in your letter, we uploaded the file of the revised manuscript. Revisions in the text are shown using red highlight for additions, and strikethrough font for deletions. The responses to the reviewer's comments are marked in red and presented following.
We would love to thank you for allowing us to resubmit a revised copy of the manuscript and we highly appreciate your time and consideration.
Sincerely,
Yan Wang.
Reviewer 2:
Q1: What specific modeling techniques were employed in the numerical simulation to assess the impact of ERSNWTP on saltwater intrusion in CRE?
Response: We are grateful for the suggestion make the following changes to clarify.
The model is divided into triangular meshes, and the north and south branches, the north and south channels, and the north and south channels are locally encrypted to better fit the shoreline of the Yangtze River estuary. The model was characterized by sensitivity analysis and determined to be most effective in the presence of Koch forces and eleven layers in the vertical direction. The initial values of the model are approximated as constants, and the initial water level and velocity fields are taken to be zero, so that deviations from the initial conditions of the flow will disappear quickly under the control of the boundary conditions. The intra-estuary is obtained by interpolating the measured data several times in the dry season. In the model, the wind field in the estuary uses the wind field hindcast data (https://www.ecmwf.int/), and the time step is 6 h. The salinity open boundary condition in the outer sea is given by the full tidal synchronization measured data, and the model linearly interpolates the boundary salinity at each computational time step, and then, according to the flow in and out of the water at the boundary, the boundary salinity is finally determined by the radial boundary condition. The salinity diffusion follows the salinity transport equation, and the ratio of the salinity horizontal diffusion coefficient to the eddy-viscosity coefficient of the water flow is considered to be 1. Finally, the model also takes into account the change of the water body density with salinity in the process of salt water intrusion, and the accuracy of the model has been improved.
In the article, two months of salinity were simulated, in which the first month was used as a stabilization time for numerical simulation, and the latter month the salinity results reached a steady state for research and analysis.
Relevant content is added to lines 125 and 134 to 138 and 158 to 160 of the article.
Q2: Could this research provide a comparative analysis of saltwater intrusion patterns before and after the ERSNWTP's water abstraction?.
Response: We are grateful for the suggestion make the following changes to clarify.
Before water withdrawal, the distance of brine intrusion was greater at high tide than at low tide, and the brine intrusion from the North Branch was greater than that from the South Branch. At high tide the brine enters the South Branch from the North Branch, and the phenomenon of backwater of brine from the North Branch occurs, and it is transported downstream with the tidal current at ebb tide, which is more obvious in the bottom layer than in the surface layer, and the brine intrusion in the South Branch mainly comes from the diffusion and transportation of the brine mass in the North Branch. The salinity of the South Channel is higher than that of the North Channel, and the saltwater intrusion is mainly dominated by the sinking of the North Branch saltwater and the direct intrusion of the open ocean saltwater. In the North Channel and South Channel, saltwater intrusion is dominated by direct saltwater intrusion, and the intensity of saltwater intrusion in the South Channel is significantly higher than that in the North Channel. At low tide, the contour of the upper section of the North Branch returns to the North Branch, salinity decreases, and the phenomenon of inversion of the North Branch disappears. Among them, the saltwater intrusion in the North Channel is the weakest, which is consistent with the previous results. With the weakening of the tide, the salinity contour lines of the surface layer of the North Harbor, North Trough and South Trough move seaward, and the saltwater intrusion is weakened, but the bottom layer moves landward, which shows the phenomenon of saline water wedge. The withdrawal of water from the South-to-North Water Diversion Phase I and Phase II projects resulted in an average reduction of 5% and 9% of the seaward runoff, leading to a further strengthening of the amount and distance of saline intrusion, but the overall pattern of saline intrusion did not undergo a root change because of the limited reduction in runoff.
In the case of water withdrawal in the South-to-North Water Diversion East Project, the discharged runoff volume is small, so the intensity of the salt water in the North Branch backing up into the South Branch is larger, and the average salinity of the South Branch at this time is more than 4 ‰, while in the case of no water withdrawal, due to the discharged runoff volume is larger, and the dilution of runoff on the salt water and the blocking effect of the saline water make the degree of salt water backing up in the North Branch slightly lighter compared to that, and the salinity of the South Branch is not more than 2 ‰. In the withdrawal of 500m3/s, the salinity of each water layer in the north channel and the south channel is greater than 0.45‰, and the salinity of some water layers in the north channel in CASE0 is less than 0.45‰ disappears completely in CASE1, which is unfavorable to the withdrawal of water from Qingcaosha Reservoir. The north branch is also still occupied by highly saline water, and the south branch still shows that the salinity of different water layers near the south bank of the middle and upper reaches is less than 0.45‰, while the salinity of other river sections is above 0.45‰. At the withdrawal of 1000m3/s, the impact of water transfer on saline water intrusion is further strengthened, in which the impact on the mouth of the south channel located in the Yangtze River is more obvious, and the salinity will be increased by 0.42‰ to 0.47‰.
Relevant content is added to lines 289 to 305 of the article.
Q3: How do the proposed ecological discharges intend to control saltwater intrusion, and what empirical evidence or simulations support their effectiveness?
Response: We are grateful for the suggestion make the following changes to clarify.
The ecological discharge in the article refers to the amount of estuarine discharge (Q) that can maintain the appropriate salinity of the estuary, compare Q with the runoff from the Datong hydrological gauging station minus the difference between the runoff from the South-to-North Water Diversion East Line and the difference between the runoff from the South-to-North Water Diversion East Line and the runoff from the North-to-East Water Diversion East Line (q), if Q≤q, the South-to-North Water Diversion East Line can be carried out as normal; if Q>q, the South-to-North Water Diversion East Line should be reduced or stopped, and the upstream water conservancy projects and other discharging flow can be appropriately increased. The scheduling can be changed at the right time to alleviate the impact of salt water intrusion in the downstream estuary.
The factors affecting salt water intrusion in the Yangtze River estuary are complex, related to both the natural evolution of the estuary and human factors. Solving the problem of saltwater intrusion in the Yangtze River estuary should be carried out from various aspects, including the management of the estuary itself and the application of scheduling of large-scale projects in the basin. Taking the salty tide intrusion in the Yangtze River estuary in Shanghai in February 2014 as an example, this salty tide intrusion was the longest duration in history suffered by the water source area of the Yangtze River estuary in Changhai, which caused serious impacts on the area. In order to alleviate the impact of the salty tide invasion, the Three Gorges Reservoir was built since the launch of the first "pressure salty tide" scheduling, increase the water replenishment, increase the discharge flow of 1000m3 / s. (Shanghai Yangtze River mouth water source), the water source of the Yangtze River mouth in Shanghai, the longest duration in history. (Shanghai Yangtze River mouth water source suffered the longest salty tide invasion (www.gov.cn)).
Relevant content is added to lines 461 to 468 and 471 to 474 of the article.
Q4: Further insights into modeling approaches and empirical validations would enhance the study's depth and credibility.
Response: We are grateful for the suggestion make the following changes to clarify.
The article considers the variation of water density with salinity during saltwater intrusion, and the model accuracy is improved by adding the wind field effect. The salinity diffusion follows the salinity transport equation and the ratio of the salinity horizontal diffusion coefficient to the eddy-viscosity coefficient of the water flow is 1. After sensitivity analysis, the presence of Koch's force, and eleven layers of vertical stratification are selected to study the model. The initial values of the model were approximated as constants, and the initial water level and flow velocity fields were taken as zero, which were interpolated from several measurements in the dry season within the estuary. The model validation was carried out using the correlation coefficient (CC) and the commonly used test index skill score (SS) for comprehensive evaluation, and to further improve the persuasiveness of the model validation, we added the root-mean-square error to quantify the model validation.
Relevant content is added to lines 168 and 135 and table 1 and formula 3 of the article.
We have had the language of the article touched up by native speakers of the language.

Reviewer 3 Report
Comments and Suggestions for Authors
Citing literature about occurring in the world in the introduction. The information added is laconic.
Author Response
Author's Reply to the Review Report (Reviewer 3)
Dear reviewer,
Re: Manuscript ID: sustainability-2750369 and Title: Saltwater intrusion of the Changjiang River Estuary responding to the East Route of South to North Water Transfer Project in the new period after 2003.
Thank you for your letter and the reviewers’ comments concerning our manuscript entitled “Saltwater intrusion of the Changjiang River Estuary responding to the East Route of South to North Water Transfer Project in the new period after 2003” (sustainability-2750369). Those comments are valuable and very helpful. We have read through comments carefully and have made corrections. Based on the instructions provided in your letter, we uploaded the file of the revised manuscript. Revisions in the text are shown using red highlight for additions, and strikethrough font for deletions. The responses to the reviewer's comments are marked in red and presented following.
We would love to thank you for allowing us to resubmit a revised copy of the manuscript and we highly appreciate your time and consideration.
Sincerely,
Yan Wang.
Reviewer 3:
Q1: Citing literature about occurring in the world in the introduction. The information added is laconic.
Response: We are grateful for the suggestion make the following changes to clarify.
As suggested by the reviewer, we've made changes to the introduction section.

Round 3
Reviewer 2 Report
Comments and Suggestions for Authors
Good response.